health and disease and epidemiology, ecology

early detection system, wildlife disease surveillance, wildlife morbidity, wildlife mortality, general disease surveillance, wildlife rehabilitation

**Author for correspondence:**
Terra R. Kelly
e-mail: trkelly@ucdavis.edu

†These authors contributed equally.

# Early detection of wildlife morbidity and mortality through an event-based surveillance system

Terra R. Kelly[1,†], Pranav S. Pandit[2,†], Nicole Carion[3], Devin F. Dombrowski[4], Krysta H. Rogers[3], Stella C. McMillin[3], Deana L. Clifford[3], Anthony Riberi[5], Michael H. Ziccardi[1], Erica L. Donnelly-Greenan[6] and Christine K. Johnson[2]

[1]Karen C. Drayer Wildlife Health Center and [2]EpiCenter for Disease Dynamics, One Health Institute, School of Veterinary Medicine, University of California, Davis, CA, USA
[3]Wildlife Health Laboratory, California Department of Fish and Wildlife, Rancho Cordova, CA, USA
[4]The Wild Neighbors Database Project, Middletown, CA, USA
[5]Y3TI, Lafayette, CA, USA
[6]Moss Landing Marine Laboratories/BeachCOMBERS Program, San Jose State University, CA, USA

TRK, 0000-0001-6762-7885; PSP, 0000-0001-7649-0649; CKJ, 0000-0001-6673-8743

The ability to rapidly detect and respond to wildlife morbidity and mortality events is critical for reducing threats to wildlife populations. Surveillance systems that use pre-diagnostic clinical data can contribute to the early detection of wildlife morbidities caused by a multitude of threats, including disease and anthropogenic disturbances. Here, we demonstrate proof of concept for use of a wildlife disease surveillance system, the 'Wildlife Morbidity and Mortality Event Alert System', that integrates pre-diagnostic clinical data in near real-time from a network of wildlife rehabilitation organizations, for early and enhanced detection of unusual wildlife morbidity and mortality events. The system classifies clinical pre-diagnostic data into relevant clinical classifications based on a natural language processing algorithm, generating alerts when more than the expected number of cases is recorded across the rehabilitation network. We demonstrated the effectiveness and efficiency of the system in alerting to events associated with both common and emerging diseases. Tapping into this readily available unconventional general surveillance data stream offers added value to existing wildlife disease surveillance programmes through a relatively efficient, low-cost strategy for the early detection of threats.

## 1. Introduction

Anthropogenic disturbances are increasingly threatening the health of wildlife populations, especially in areas undergoing rapid urbanization and population growth [1–3]. These disturbances are contributing to a wide range of threats, including habitat fragmentation, invasive species introductions, pollution and disease emergence [4–6]. Infectious diseases contribute to biodiversity loss [7], and there is mounting evidence that pollution and emerging pathogens have devastating impacts on wildlife populations around the world (e.g. amphibian chytridiomycosis, white-nose syndrome in bats and domoic acid intoxication in marine animals) [5,8,9].

These same disturbances drive emerging disease risk in people [6,10–12]. The increasing incidence of emerging infectious diseases, the majority of which originate in wildlife [6], has become one of the most prescient challenges facing human and animal health [10,13,14]. Notable examples include the emergence and spread of Ebola virus [15], Nipah virus [16] and most recently SARS-CoV-2 [17].

With increasing awareness of the impacts of emerging diseases on both humans and animals and the importance of wild animals as hosts and/or reservoirs of zoonotic pathogens, there is a greater recognition of the need for disease surveillance in free-ranging wildlife. Reports of unusual illnesses or deaths in wildlife populations may serve as the first alert of an emerging health threat [18]. For example, mortality in crows (*Corvus* species) and exotic birds at a zoologic park in New York provided early warning of the emergence of West Nile Virus (WNV) in the United States (US) in 1999. Avian deaths were a critical precursor to identifying WNV as the cause of the associated human encephalitis outbreak [19]. As crow mortalities were occurring prior to the onset of human WNV cases, dead bird surveillance served as a sensitive indicator for WNV activity and the risk of infection for humans [20].

Surveillance strategies focusing on early detection are important for quickly identifying and mitigating threats as they emerge [18]. Targeted surveillance, which focuses on a particular pathogen or contaminant, is valuable for surveilling specific disease agents in wildlife [21–23]. However, the implementation of targeted surveillance is costly and time-consuming for surveilling a wide range of threats across multiple species [20,21,23]. Targeted surveillance strategies can also miss trends signifying an emerging threat that is outside the systems' targets. Therefore, wildlife disease surveillance programmes typically also incorporate general disease surveillance, a strategy for detecting sick and dead animals in the wild and identifying causes of morbidity and mortality [23]. General disease surveillance is not limited to a few species or pathogens, rather it covers a broad range of wildlife and causes of illness and death [23].

General surveillance systems, especially systems that use existing pre-diagnostic health information or syndromic data can facilitate early detection of health threats through the identification of unusual disease clusters early before diagnoses are confirmed and officially reported [18,24]. While syndromic systems do not track verified events, they can provide a valuable complement to targeted surveillance through alerting to anomalous wildlife morbidity events, thereby enhancing situational awareness and increasing opportunities for early detection and response. Syndromic classification of wildlife mortalities based on pre-diagnostic post-mortem examination findings has been highlighted as a rapid, reliable and relatively inexpensive strategy for disease surveillance [25]. Using pre-diagnostic clinical wildlife health data generated through physical examination findings is a novel strategy that offers an even more efficient approach to syndromic surveillance as the data is entered in near real-time upon admission of animals to the centres. It is also practical in settings where it is not feasible to classify high numbers of cases based on their pathologic profiles through post-mortem examinations. By using existing clinical wildlife health data, this approach can also provide a relatively inexpensive means to bolster disease surveillance programmes [26].

In North America, wildlife agencies conduct targeted disease surveillance for several endemic and recently emerging wildlife diseases of importance (e.g. avian influenza, rabies, white-nose syndrome, snake fungal disease and bovine tuberculosis) [27–30]. These agencies also investigate reports of unusual morbidity and mortality events in order to detect anomalies outside of surveillance targets, including new species or geographical areas affected by known diseases or new diseases as they emerge.

With increasing focus on the importance of early detection and the need for innovative, cost-effective strategies, several new approaches and tools have been developed as a complement to conventional surveillance systems. For example, citizen science has increasingly been used in North America for surveilling diseases causing characteristic clinical signs, such as avian pox or finch conjunctivitis [20,31]. Agencies have also implemented harvest-based disease surveillance as a practical, cost-effective strategy (e.g. chronic wasting disease in cervids and *Brucella* spp. in coyotes) [32,33].

Wildlife rehabilitation organizations are also increasingly recognized for their potential to contribute to disease surveillance, including diseases of importance to domestic animal and human health [34–36] as well as emerging threats [31,34,37]. Until recently, platforms for sharing information among organizations have been lacking, resulting in missed opportunities to detect unusual events when sick animals are brought to multiple, uncoordinated centres across a region. In addition, efforts to use rehabilitation data to contribute to disease surveillance have previously focused on verified diagnostic data rather than clinical pre-diagnostic data, limiting the information's use to contribute to early detection. When linked through a formal network, these organizations have the potential to assimilate and share large amounts of clinical data in near real-time, collectively representing a highly valuable and under-used resource that can be used to complement existing surveillance efforts for early and enhanced detection [21,31,34,35,38–42]. In this study, we aimed to develop and pilot an online surveillance system that integrates pre-diagnostic clinical health data entered in near real-time by a network of wildlife rehabilitation organizations to facilitate early and enhanced detection of wildlife morbidity and mortality events.

## 2. Methods

In 2012, The Wildlife Neighbors Database Project developed the Wildlife Rehabilitation Medical Database (WRMD; https://www.wrmd.org/), a free online database designed for wildlife rehabilitation organizations to compile, analyse and archive standardized patient data. WRMD currently contains over 2 million wildlife patient records, and data are entered by 950+ organizations across 48 US states and 19 countries. To build on the capabilities of this database, we developed a web-based surveillance platform, the 'Wildlife Morbidity and Mortality Event Alert System' (WMME Alert System), that runs in parallel with WRMD to rapidly detect wildlife morbidity and mortality events. The platform integrates data entered in WRMD in near real-time by a network of 30 wildlife rehabilitation organizations across California (figure 1). Data for each case includes a unique identifying number, species, sex, age class, location found, circumstances of admission, initial examination findings and ultimately the diagnosis if determined. Personal identifiable information of rescuers is excluded. Aggregated data displayed through interactive tabular, graphical, and spatial dashboards in the WMME Alert System are accessible to the network of partner wildlife rehabilitation organizations and the Wildlife Health Laboratory (WHL) of the California Department of Fish and Wildlife, which leads the state's wildlife disease surveillance efforts.

Wildlife data (219 767 case records) collected between 1 January, 2013 and 31 December, 2018 were extracted from WRMD to establish thresholds for triggering alerts in the WMME Alert System (electronic supplementary material, Data File S1). Alerts to anomalous events are generated when the number of cases exceeds pre-defined thresholds for the number of admissions

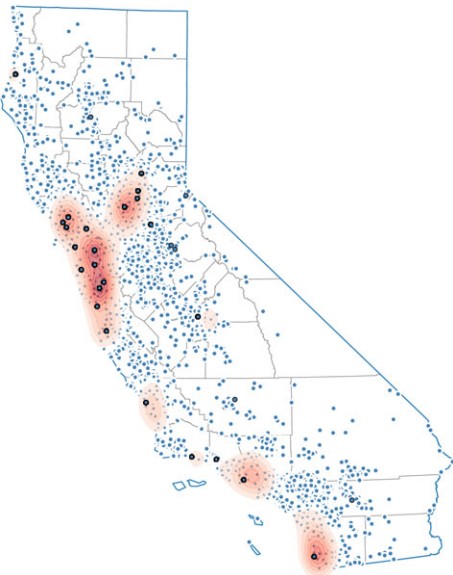

**Figure 1.** Locations of cases (smaller blue dots) presenting to a network of wildlife rehabilitation organizations (bigger blue dots) participating in the Wildlife Morbidity and Mortality Event Alert System in California, from 2013 to 2018. Red region shows areas with high kernel density of cases.

for a given species and for the number of admissions for a given species presenting with a specific clinical classification (e.g. neurologic disease) based on physical examination findings.

## (a) Clinical classification of cases and development of alert thresholds

To establish the alert thresholds specific to clinical classifications, a supervised machine learning algorithm was used to assign one of 12 pre-diagnostic clinical classifications to each case in the dataset (table 1). A total of 3081 cases were randomly chosen from the data as a training dataset (electronic supplementary material, Data File S2). Each of these records was assigned one of 12 clinical classifications based on data recorded for 'reasons for admission', 'initial physical examination', and 'preliminary diagnosis' by a wildlife veterinarian (T.R.K). Twenty per cent of the training case records ($n = 600$ entries) were randomly chosen as a test dataset for model validation. The remaining dataset was used for cross-validation. Specifically, a 'bag of words' approach [43] was used to extract predictive feature data and generate a list of vocabulary words from text entered in the 'reasons for admission', 'initial physical examination', and 'preliminary diagnosis' fields in WRMD. After grouping together, the inflected forms of words in these data fields were tokenized to produce a sparse matrix of feature data. Using tokenized vector data, a support vector classification (SVC) algorithm was trained to predict the clinical classification for each case. The model was parameterized on the training data using 10-fold cross-validation. To identify the best hyperparameters of the SVC classification algorithm, a grid search was implemented with the cross-validation process covering a wide range of model parameters (electronic supplementary material, table S1). Best performing model parameters were chosen based on accuracy, precision and recall. Eventually, the best performing model was tested for accuracy, precision and recall on the independent test dataset. Modelling was implemented in Python using the Scikit-learn Machine Learning Package [44].

Anomalies in admissions for each species and for each species/clinical classification combination were identified in the dataset using estimates of the rolling mean and rolling standard deviation derived through time-series analyses. Anomalies were identified as admissions exceeding the thresholds. Thresholds were defined as two times the standard deviation above the rolling mean (rolling mean + [2 × rolling standard deviation]). Thresholds account for seasonality and trends in weekly times series for a given taxonomic group and clinical classification (equation (2.1)). Alerts are generated when there are higher than expected numbers of admissions of a particular species and taxa group with a specific clinical classification.

$$p(n)_t = \{1 \; if \; Cases_{t,sp,c}$$
$$> MA_{t,sp,c} + 2SD(MA_{t,sp,c}), \; 0 \; otherwise \;, \qquad (2.1)$$

where $MA_{t,sp,c}$ is the moving average of taxonomic group $sp$ cases presenting with $c$ pre-diagnostic clinical classification at time $t$ and $SD(MA_{t,sp,c})$ is the moving standard deviation.

## (b) Use for detection of a wide range of wildlife morbidity and mortality events

We evaluated the system's geographical coverage as well as the diversity of wildlife species represented. To evaluate the spatial coverage, we assessed the distribution of cases in the dataset by estimating spatial kernel density. Reported geolocations of cases were used to fit a 'biweight' kernel distribution with a grid size of 100 using the 'geoplot' package in Python (https://github.com/ResidentMario/geoplot). The densities were visualized and mapped along with the geolocations of cases (geocoded according to the addresses where animals were found) and organizations.

Over one year, we conducted in-depth investigations of morbidity and mortality events involving a range of wild avian and mammalian species that were triggered by alerts in the system. Investigations were performed by the WHL in collaboration with the network of organizations. The investigations included full post-mortem examinations and ancillary diagnostic testing to determine the causes of morbidity/mortality for each event. Examinations and diagnostic assays were performed at the WHL (Rancho Cordova), California Animal Health and Food Safety Laboratory (Davis, CA), US Geological Survey National Wildlife Health Center (Madison, WI) and other specialized laboratories. Once verified with a laboratory diagnosis, information on morbidity and mortality events fed into a national wildlife disease surveillance system—the USGS NWHC Wildlife Health Information Sharing Partnership—event reporting system (WHISPers). WHISPers is a publicly available web-based data repository for sharing information on wildlife health events with the goal of providing managers and the public with timely, accurate information on wildlife health threats [45].

## (c) External validation

For validation of the WMME Alert System, we performed time-series analyses to compare trends in data generated through the system to an independent data source. Specifically, we compared data on marine bird admissions to data on stranded marine birds collected by a group of volunteers called BeachCOMBERS (BC). The BC data was systematically collected through standardized beach surveys conducted monthly in southern California to record data on stranded marine birds and mammals (electronic supplementary material, Data File S3). We used a subset of the BC data recorded from January 2013 to December 2018 for comparison to marine bird data arising from cases presenting to wildlife organizations in southern California through the WMME Alert System. Autocorrelations of the time series and the augmented Dickey-Fuller test were used to evaluate the stationarity of the time series. The Granger test of causality was used to investigate whether there was an association between marine bird admissions in the system and reports of stranded birds in the BC data. In addition,

**Table 1.** Definitions of pre-diagnostic clinical classifications for categorizing cases.

| clinical classification | definition |
| --- | --- |
| neurological disease | conditions affecting the central and peripheral nervous systems |
| respiratory disease | conditions affecting the organs and tissues that make gas exchange possible and includes conditions of the upper respiratory tract, trachea, bronchi, bronchioles, alveoli, pleura and pleural cavity |
| gastrointestinal disease | conditions affecting the gastrointestinal tract, namely the oesophagus, stomach, small intestine, large intestine and rectum, and the accessory organs of digestion, the liver, gallbladder and pancreas |
| haematological disease | conditions affecting the red blood cells, white blood cells, platelets, blood vessels, bone marrow, lymph nodes, spleen and the proteins involved in bleeding and clotting |
| dermatological disease | conditions affecting the skin, fur and feathers |
| ocular disease | conditions affecting any of the eye components such as cornea, iris, pupil, optic nerve, lens, retina, macula, choroid, conjunctiva or the vitreous |
| nutritional disease | pertaining to any disease resulting from an alteration in the processes involved in taking nutrients into the body and assimilating and using them or from deficiencies or excesses of specific feed nutrients |
| petrochemical exposure | exposure to petrochemical (oil, grease, paint, etc.) causing external contamination of the animal and/or leading to ingestion of the chemical |
| physical injury | injury caused by trauma from an external force (mechanical, thermal, electrical, chemical) |
| stranded | referring to events leading to single or multiple animals that are cut off from their natural habitat and cannot be returned unassisted. Often caused by altered behaviour such as marine bird stranding |
| orphaned | displaced healthy or injured young animal, still dependant on parental care for survival |
| nonspecific | not assignable to a particular category or classification |

the cross-correlation function was used to explore the relationship between the two time series and to identify lags in one series relative to the other. Finally, an autoregressive integrated moving average model (ARIMAX) with the WMME Alert System marine bird data as an external variable was fitted to evaluate the association between the WMME Alert System data and BC stranded bird data. Data from 2013 to 2017 was used for training the ARIMAX model, and 2018 data was used for validation. To identify the parameters of the ARIMAX model, the best-fitting model was selected using the auto.-arima function in the forecast library of R. The p,d,q parameters of the model were selected based on the best model from the auto.ar-ima function. Following identification of the p,d,q parameters, seasonality variables (P,D,Q) were included and various values were tested (electronic supplementary material, table S3). From these, three models with the least Akaike information criterion corrected for small sample size (AICc) were selected and used for forecasting. The accuracy of the forecast for all three models was estimated using root mean squared error (RMSE) and mean absolute percentage error (MAPE). The model forecasting data most similar (based on RMSE and MPAE) to that of the observed data was selected as the final model.

# 3. Results

The WRMD dataset included records from 453 different species among 27 taxonomic orders, illustrating the high diversity of species represented in the system. However, 43 species comprised 80% of the total data and species commonly found in human-dominated landscapes (e.g. northern raccoon (*Procyon lotor*) and American crow (*Corvus brachyrhynchos*)) were prevalent. Cases originated from all counties in California with the highest densities of admissions in urban/semi-urban areas along the coast and in the Central Valley (figure 1) with clustering around the wildlife rehabilitation organizations.

## (a) Clinical classification of cases

The best-fitting SVC model predicted the pre-diagnostic clinical classifications in the holdout dataset (test dataset) with an overall accuracy of 83% and precision of 0.84 (recall = 0.83, F1-score = 0.83, $n = 617$). The SVC model was very accurate (92% accuracy) in classifying cases with physical injury (precision = 0.78, recall = 0.91, F1-score = 0.84, $n = 191$). However, the lower precision for this category illustrates that some cases from other clinical classifications were misclassified as physical injury (figure 2; electronic supplementary material, table S2). Specifically, 15% of nutritional and respiratory disease cases, 14% of skin, ocular and gastrointestinal disease cases and 13% of neurological disease cases were falsely identified as physical injury. Misclassification also occurred for some cases of neurological disease, with 15% and 14% of animals presenting with respiratory and gastrointestinal disease, respectively, categorized as cases of neurological disease (accuracy = 83%, precision = 0.78, recall = 0.83, F1-score = 0.80, $n = 123$). On the other hand, the model demonstrated perfect precision for classifying petrochemical exposure cases with no false-negative predictions (precision = 1.0, accuracy = 91%, recall = 0.91, F1-score = 0.95, $n = 22$). Receiver operating curves for clinical classifications are shown in the electronic supplementary material, figure S1.

## (b) Use for detection of a wide range of wildlife morbidity and mortality events

Over the one-year pilot period, the WMME Alert System detected several anomalies that, upon investigation, were found to be caused by common causes of wildlife morbidity and mortality in California as well as emerging health threats

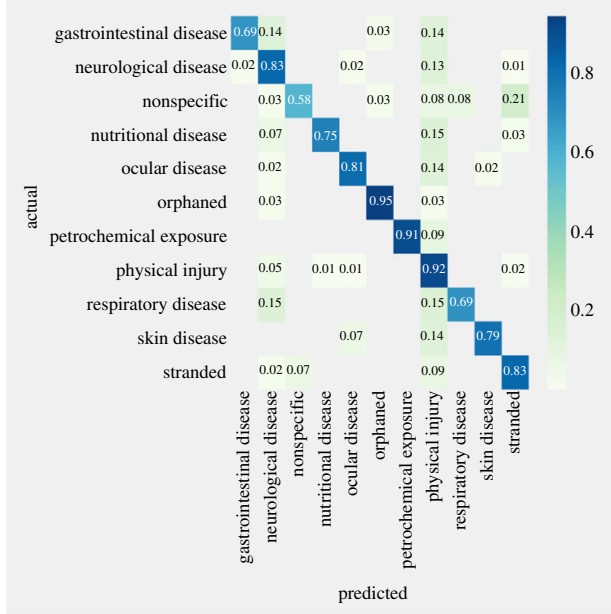

**Figure 2.** Confusion matrix showing proportion of cases correctly classified and misclassified by the support vector classifier model (*x*-axis) using expert based classification (*y*-axis) of cases into clinical classifications.

(table 2). Here, we describe four key investigations of events after consistent weekly, bi-weekly, and monthly alerts in the system.

In the late spring of 2016, a large influx of marine birds along the central and southern California coast was detected through weekly alerts (figure 3a). Several western grebes (*Aechmophorus occidentalis*) and to a lesser extent Clarke's grebes (*Aechmorphorus clarkia*) and eared grebes (*Podiceps nigricollis*) were involved in the event. The birds were found to be emaciated upon post-mortem examination. Similarly, the system detected an unusual event in marine birds in southern California in April 2017 (figure 3a) with weekly, bi-weekly and monthly alerts generated during the event. Upon investigation, several marine bird species were found to be affected, including California brown pelicans (*Pelecanus occidentalis*), Pacific loons (*Gavia pacifica*), red-throated loons (*Gavia stellate*), common murres (*Uria aalge*), western grebes, Clark's grebes and Brandt's cormorants (*Phalacrocorax penicillatus*). The birds presented with neurological disease, including head twitching and whole-body tremors. Post-mortem examinations and diagnostic testing revealed domoic acid intoxication as the cause of death.

The WMME Alert System also alerted to cases of neurological disease in doves that were associated with the northward spread of pigeon paramyxovirus type 1 (PPMV-1), an emerging virus in California. Starting in the late summer of 2016, there were increased admissions of invasive Eurasian collared doves (*Streptopelia decaocto*) (figure 3b) in central and northern California. Affected doves displayed neurological signs including abnormal twisting/tilting of the neck and paralysis. Encephalitis and kidney disease were identified on post-mortem examination. Polymerase chain reaction (PCR) and sequencing confirmed the presence of pigeon paramyxovirus-1, the first detection of the virus emerging in Eurasian collared doves in this region of California [46]. Cases continued with another event occurring in the late summer/early autumn of 2017.

The WMME Alert System also detected increased admissions associated with an outbreak of neurological disease in rock pigeons (*Columba livia*) in the San Francisco Bay area

in late winter/early spring of 2017 (figure 3c) [35]. Meningoencephalitis and protozoal organisms were observed on post-mortem examination. Pan-*Sarcocystis* PCR identified *Sarcocystis calchasi* group infections in several of the pigeons and sequences detected in eight cases had 100% homology with *S. calchasi* [35]. This event demonstrated the emergence of this parasite in free-ranging birds in California and highlighted the importance of increased surveillance in susceptible native columbids.

Seasonal conjunctivitis events in finches were also detected by this system, including events in the spring of 2016 and early months of 2017 (figure 3d). Several finch species were affected with conjunctivitis, with some cases also exhibiting upper respiratory disease. *Mycoplasma gallisepticum* was confirmed by real-time PCR (RT-PCR) as the cause of conjunctivitis among tested finches. Infection in American goldfinches expanded the known host range of *Mycoplasma* spp. conjunctivitis in California [47].

Along with these aforementioned investigations, alerts generated through the WMME Alert System demonstrated the wide use of the system in detecting anomalies in single species as well as in groups of related species (e.g. nutritional disease in loons and grebes (electronic supplementary material, figure S2) and double-crested cormorants (electronic supplementary material, figure S2)). Similarly, the system's ability to detect events associated with endemic and emerging pathogens causing neurological diseases in birds, such as WNV, pigeon paramyxovirus-1, and *S. calchasi* was illustrated through investigations of anomalies in neurological cases detected in Cooper's hawks (figure 4a) and species from the Columbidae family (figure 4b). Anomalies associated with toxicities were also captured and tracked using the system as evidenced by the detection of petrochemical exposure in marine birds (electronic supplementary material, figure S2). Various investigations in mammals were also triggered owing to system alerts. For example, canine distemper virus (CDV) infection and bromethelin intoxication were associated with anomalies in raccoon and skunk admissions (figure 4c). Not surprisingly, the system was also able to track trends in admissions of animals associated with physical injury (e.g. vehicular trauma in deer; figure 4d), a common circumstance of admission to rehabilitation organizations. Events in rare species, such as increased cases of neurological disease in golden eagles (*Aquila chrysaetos*) (electronic supplementary material, figure S2), can also be monitored in the system. However, the specificity for alerts in these species tends to be lower given the relatively fewer numbers presenting to organizations. Exploring trends at the taxonomic family level (i.e. *Accipitridae* family), in addition to the species level, could provide additional insights into the health of threatened and rare species (electronic supplementary material, figure S2).

## (c) External validation

The time series (BC and WMME Alert System datasets) showed similar trends over the 5 years (figure 5) and were found to be stationary (augmented Dickey-Fuller test; BC $p = 0.001$, WMME Alert System $p = 0.002$). The cross-correlation function showed that the number of cases in the WMME Alert System during the previous month was the most influential on the number of stranded birds recorded by BC in a given month (electronic supplementary material figure S3). In addition, the Granger test of causality at a lag of one month

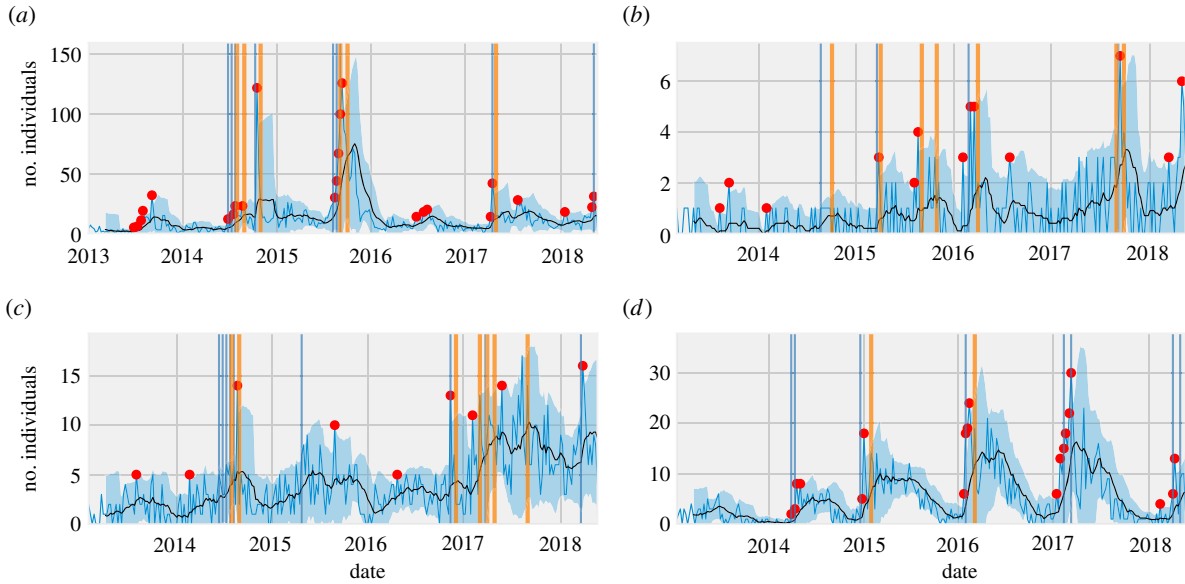

**Figure 3.** Alerts generated for four wildlife disease investigations in California. Weekly alerts are represented by red dots and bi-weekly alerts and monthly alerts are represented by blue and orange vertical lines, respectively. The timeline of the weekly number of cases is presented by a sky-blue line and the black line represents rolling mean with window of 10 weeks. Blue-shaded region shows twice the rolling standard deviation from the rolling mean. (*a*) Strandings in marine birds, (*b*) neurological disease in Eurasian collared doves, (*c*) neurological disease in rock pigeons, and (*d*) ocular disease in finches.

**Table 2.** Examples of wildlife morbidity and mortality events caused by endemic and emerging threats appearing as alerts in the WMME Alert System.

| common/endemic threats | | emerging threats | |
|---|---|---|---|
| species/taxa | aetiology | species/taxa | aetiology |
| finches | *Mycoplasma* spp. conjunctivitis | Eurasian collared doves | pigeon paramyxovirus-1 |
| Cooper's hawks | WNV | rock pigeons | *Sarcocystis calchasi* |
| mourning doves | trichomoniasis | | |
| raccoons | bromethelin intoxication, CDV | | |
| turkey vultures | lead intoxication | | |
| marine birds | domoic acid intoxication, starvation, petroleum contamination | | |

was significant ($p = 0.01$) indicating that the WMME Alert System data had incremental power to forecast the number of stranded birds in the BC data.

The auto.arima function identified the ARIMAX model with parameters $p = 2$ (number of autoregressive terms), $d = 0$ (number of non-seasonal differences needed for stationarity) and $q = 0$ (number of lagged forecast errors in the predictions equation) as the best-fitting model (ARIMAX (2, 0, 0) errors, AICc of 715.56, electronic supplementary material table S3). Following testing of seasonality parameters (P, D, Q), three models with combinations of (0,1,0), (2,2,1) and (2,2,0) were selected based on AICc (electronic supplementary material, table S4), where P is the number of seasonal autoregressive terms, D is the number of seasonal differences and Q is the number of seasonal moving average terms. Among the three best-fitting models, ARIMAX (2,0,0) (0,1,0) [12] errors, showed the least RMSE (98.49) and MAPE (37.23) and predicted the BC data similar to the observed data (figure 5; electronic supplementary material, table S5). This model also revealed that WMME Alert System data with a lag of first order was significantly associated with the number of stranded marine birds in the BC dataset ($p > 0.001$; electronic

supplementary material, table S6). Taken together, the time series analyses suggest that marine bird admissions in the WMME Alert System precede documentation of strandings using existing survey methods by approximately one month and therefore contribute to early detection of these events.

## 4. Discussion

We demonstrate the use of an online surveillance system integrating clinical pre-diagnostic data from a network of wildlife rehabilitation organizations to facilitate early and enhanced detection of wildlife morbidity and mortality events in California. The WMME Alert System represented a wide range of wildlife species and covered a broad area across the state given the extensive reach of the network of participating organizations. However, the majority of animals admitted were common species frequently found in human-dominated landscapes. In addition, although cases originated from all counties, most admissions originated from urban and semi-urban areas along the coast and as expected, the highest densities of cases clustered around the rehabilitation

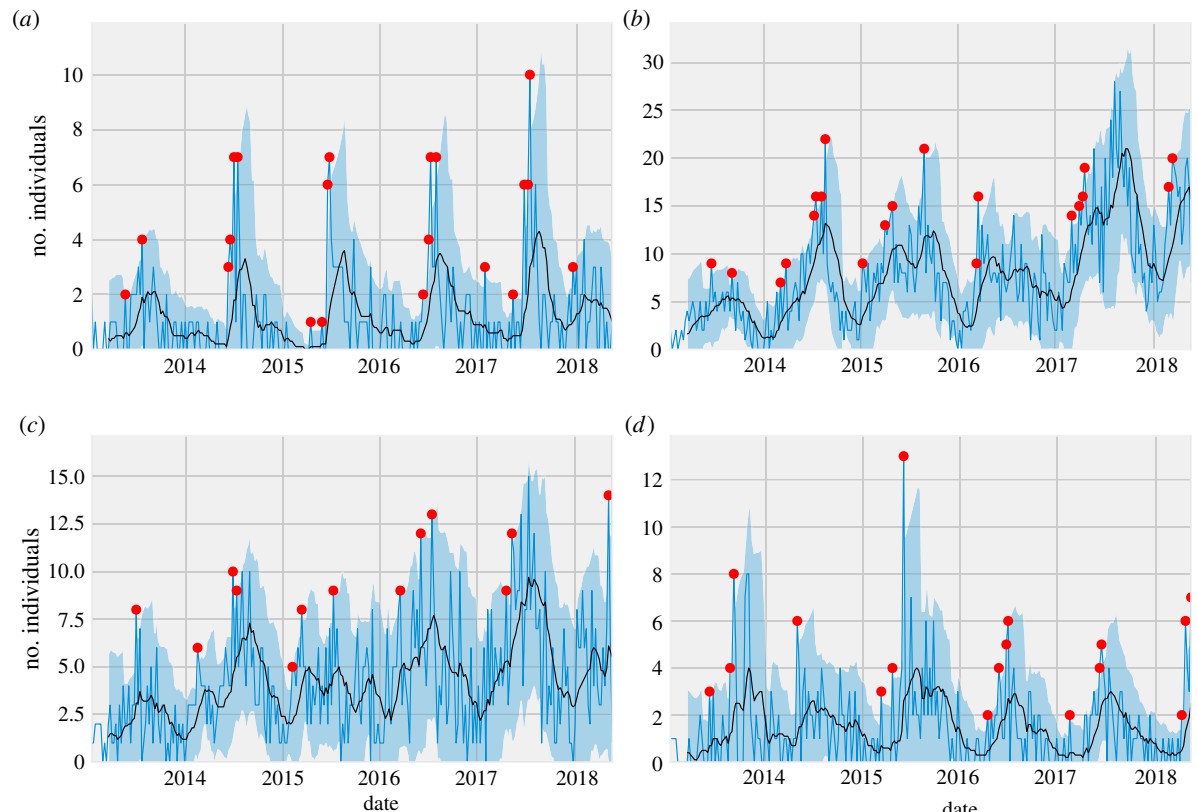

**Figure 4.** Alerts generated for various species and groups of species for different clinical classifications. Blue line shows weekly number of cases presenting to rehabilitation organizations across California from 2013 to 2018. Black line represents rolling mean with window of 10 weeks. Blue-shaded region shows twice the rolling standard deviation from the rolling mean. Red dots represent temporal anomalies for weekly number of cases. (a) Neurological disease in Cooper's hawks, (b) neurological disease in Columbids, (c) neurological disease in raccoons and skunks, and (d) physical injury in deer.

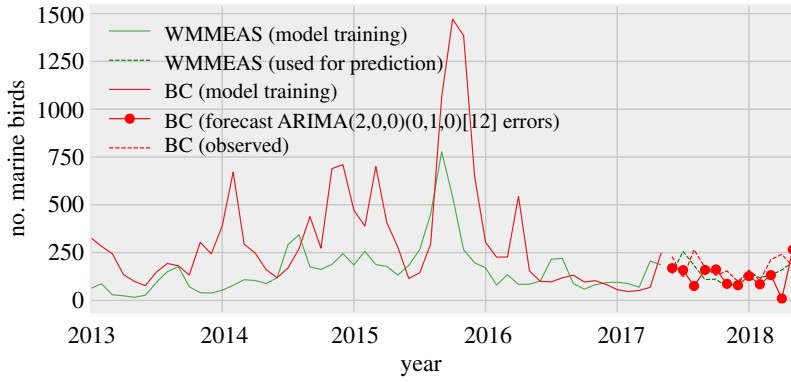

**Figure 5.** Number of monthly reported strandings (BC data) in southern California and number of admissions (WMMEAS data) in nearby rehabilitation organizations. Forecast of BC data using ARIMAX model with WMMEAS as external regressor.

organizations. This finding reflects the inherent reporting bias of wildlife disease surveillance systems that rely on the public for the initial detection of cases. However, this system, together with other general disease surveillance efforts (i.e. citizen wildlife mortality event reporting), are important complements to targeted surveillance efforts in California (e.g. chronic wasting disease in cervids and white-nose syndrome in bats) through efficient monitoring for emerging threats across a broad range of species and geographies [40], especially species in disturbed environments [35]. The information generated through this system adds value to other general surveillance strategies through its ability to rapidly and efficiently detect threats that lead to illness

and death in wild animals but do not necessarily result in conspicuous mortality events that would be detected through citizen reporting streams.

As front-line responders for injured and sick wild animals, wildlife rehabilitation organizations are well poised to detect index cases of emerging wildlife health threats [34,36]. Enhanced capacity to quickly identify unusual cases or patterns is becoming more important with increasing anthropogenic pressures causing unforeseen threats (emerging infectious diseases and environmental pollutants) that can result in population declines [48] and endangerment of common species [49]. As emerging threats become more commonplace, there is a greater need for wildlife disease

surveillance programmes that extend beyond tracking only known hazards [50] and have the capacity to rapidly detect small isolated events.

This surveillance application was effective in detecting anomalous patterns of admissions across the network of organizations that upon investigation were determined to be the result of both common and emerging health threats. Common health threats such as *Mycoplasma* spp. conjunctivitis in songbirds, trichomoniasis in doves, CDV in raccoons and petroleum contamination of marine birds were detected with support from this system's alerts, illustrating its use for monitoring trends in these diseases over time. The system also detected events that upon investigation were identified to be the result of emerging diseases in peridomestic and/ or invasive species that present a threat to native wildlife [35,46]. Detecting anomalies in admissions associated with emerging diseases in wildlife illustrates this system's capacity to detect anomalous events associated with a novel threat.

Passive data streams have value in that they offer a broad sweep for identifying threats, including emerging diseases, that would be missed by targeted efforts. The broad clinical classifications and flexibility to assess trends in single species or taxa in this system offer a sensitive and rapid method for detecting anomalies. Overall, the model used to predict the clinical classifications demonstrated high accuracy. Misclassification of cases occurred primarily owing to similarities in vocabulary in the reasons for admission and initial physical examination fields across multiple classifications. For example, birds presenting with physical injury (i.e. head trauma) were sometimes misclassified as neurological disease cases owing to similar verbiage across those two classifications. This type of misclassification can be reduced through the inclusion of multiple clinical classifications. A multi-output classification system, in which a single case can be assigned two or more clinical classifications, is currently under development in the system. In addition, even though the system's specificity was lower for detecting events in rare species, alerts involving a small number of individuals of a threatened or endangered species signifying a potential anomalous event may be worthy of investigation. Monitoring for alerts in sympatric species and/or in related but more common species at a taxa level might also cue investigators into a common threat that could impact the health of threatened and endangered species. The precision of the model will also improve over time as the system becomes more populated with data.

Our external evaluation of the system illustrated its capacity to support the early detection of anomalous events. Specifically, the time series analyses revealed the system's ability to detect anomalies in stranded marine birds presenting to rehabilitation organizations earlier than standard active systems using data generated through existing surveys. Early detection of cases in this context could be owing in part to the near real-time integration of data in the surveillance system as compared to the monthly survey data collection on stranded birds.

The effectiveness of this type of system is linked to timely and accurate data entry by rehabilitation organizations. We found that most organizations entered data daily as part of their standard patient care. To improve on this system, an increased focus on standardization of data entry by organizations is ongoing. Greater standardization through autocomplete text features with standardized terminology and training of staff on key terminology will reduce errors and inconsistencies across users. This will further promote the use of this data for general surveillance, situational awareness, prioritization of targeted surveillance efforts and research on health threats.

## 5. Conclusion

We provide proof of concept for using pre-diagnostic clinical data assimilated from a network of wildlife rehabilitation organizations to contribute to early and enhanced detection of wildlife morbidity and mortality events. The WMME Alert System serves as a model for a relatively efficient, inexpensive system that capitalizes on existing data sources to augment surveillance and monitoring efforts and promote situational awareness. In addition, the platform or its framework provides an effective strategy for early detection of anomalous events across broad species, geographies and threats and has the capacity to be scaled up, adapted and applied in other regions or contexts, including where diagnostic capacity is limited. It serves as a valuable tool for assisting with early detection of and alerting to emerging diseases of wildlife as well as threats to domestic animal and human health (e.g. harmful algal blooms). The potential exists to expand the network to additional organizations involved in wildlife care and research and to create separate networks in other regions around the world given the current reach of WRMD.

Data accessibility. The data and code (Python and R) used for developing models and generating figures needed to replicate and evaluate these analyses are provided at https://zenodo.org/record/4920052. The datasets supporting this article have also been uploaded as part of the electronic supplementary material.

Authors' contributions. T.R.K.: conceptualization, data curation, formal analysis, funding acquisition, investigation, methodology, project administration, supervision, writing—original draft, writing—review and editing; P.S.P.: data curation, formal analysis, investigation, methodology, visualization, writing—original draft, writing—review and editing; N.C.: conceptualization, funding acquisition, investigation, project administration, supervision, writing—original draft, writing—review and editing; D.F.D.: conceptualization, data curation, formal analysis, funding acquisition, methodology, software, visualization, writing—original draft, writing—review and editing; K.H.R.: conceptualization, investigation, methodology, writing—original draft, writing—review and editing; S.C.M.: conceptualization, investigation, methodology, writing—original draft, writing—review and editing; D.L.C.: conceptualization, investigation, methodology, writing—original draft, writing—review and editing; A.R.: conceptualization, formal analysis, methodology, visualization, writing—review and editing; E.L.D.: conceptualization, methodology, resources, writing—original draft, writing—review and editing; M.H.Z.: conceptualization, methodology, writing—original draft, writing—review and editing; C.K.J.: conceptualization, methodology, writing—original draft, writing—review and editing. All authors gave final approval for publication and agreed to be held accountable for the work performed therein.

Competing interests. We declare we have no competing interests.

Funding. This work was funded through a State Wildlife Grant (Agreement no. F14AF00639).

Acknowledgements. We thank all of the participating organizations in California: Bird Ally X, Bird Rescue Center, California Wildlife Center, California Living Museum, Fresno Wildlife Rescue and Rehabilitation Service, Fawn Rescue, Gold Country Wildlife Rescue, Injured and Orphaned Wildlife, Lindsay Wildlife, Lake Tahoe Wildlife Care, Inc., Mother Lode Wildlife Care, Native Songbird Care and Conservation, Native Animal Rescue, Ojai Raptor Center, Pacific Wildlife Care, Peninsula Humane Society and SPCA, Project Wildlife, Rose Wolf Wildlife Rescue and Rehabilitation, Sonoma County Wildlife Rescue, SPCA for Monterey County, Sulphur Creek Nature Center, Santa Barbara Wildlife

Care Network, Tehama Wildlife Care, The Living Desert, Wildlife Rehabilitation and Release, Wildlife Center of Silicon Valley, Wildlife Rescue Center of Napa County, WildCare, Wildlife Care Association of Sacramento. We are also grateful to Steve Torres for his intellectual contributions, Nicholas Shirkey for mortality investigation support, BeachCOMBERS Program for sharing data on marine bird strandings, and to Corrine Gibble for her input on external validation.

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
