## [Peer Review File · Proceedings of the Royal Society B: Biological Sciences]

Review History

RSPB-2020-2317.R0 (Original submission)

Review form: Reviewer 1

Recommendation

Accept with minor revision (please list in comments)

Scientific importance: Is the manuscript an original and important contribution to its field?

Excellent

General interest: Is the paper of sufficient general interest?

Good

Quality of the paper: Is the overall quality of the paper suitable?

Excellent

Is the length of the paper justified?

Yes

Should the paper be seen by a specialist statistical reviewer?

Yes

Do you have any concerns about statistical analyses in this paper? If so, please specify them explicitly in your report.

No

It is a condition of publication that authors make their supporting data, code and materials available - either as supplementary material or hosted in an external repository. Please rate, if applicable, the supporting data on the following criteria.

Is it accessible?

Yes

Is it clear?

Yes

Is it adequate?

Yes

Do you have any ethical concerns with this paper?

Yes

Comments to the Author

The authors present a proof of concept paper on use of wildlife rehabilitation centers as an early monitoring system for disease outbreaks. This can be a highly useful system for disease outbreaks and these centers are terribly underused for this purpose. There were minor edits that I made in the document, but my main concern is that the authors fail to demonstrate how truly important this system could be in detecting novel infectious or non-infectious pathogens. The examples in the paper utilizing existing pathogens on the landscape are important, but hardly demonstrate the usefulness of this system to an outside user. Epidemiologists would appreciate the opportunity to track existing diseases using this system, but the authors contend that it's also useful for novel pathogens. Thus, it would be nice if they could identify an example in the historic data set and include that example as well.

Review form: Reviewer 2

Recommendation

Major revision is needed (please make suggestions in comments)

Scientific importance: Is the manuscript an original and important contribution to its field?

Good

General interest: Is the paper of sufficient general interest?

Acceptable

Quality of the paper: Is the overall quality of the paper suitable?

Excellent

Is the length of the paper justified?

Yes

Should the paper be seen by a specialist statistical reviewer?

Yes

Do you have any concerns about statistical analyses in this paper? If so, please specify them explicitly in your report.

No

It is a condition of publication that authors make their supporting data, code and materials available - either as supplementary material or hosted in an external repository. Please rate, if applicable, the supporting data on the following criteria.

Is it accessible?

Yes

Is it clear?

Yes

Is it adequate?

Yes

Do you have any ethical concerns with this paper?

No

Comments to the Author

Main comments:

This is a article on a wildlife disease syndromic surveillance system using data from wildlife rehabilitators on a free online database that used machine learning methods to detect early wildlife mortality events in avian species. This WRMD seems to be a fantastic resource.

Your use and analyses of data on early detection of wild bird mortality events is important and novel for this field of wildlife diseases. The audience, which would be wildlife disease managers, state, provincial, and other types of government agencies dealing with wildlife surveillance, may miss this article in Proc B, but would more readily read the Journal of Wildlife Diseases, Wildlife Management, wildlife society journals, applied conservation journals, etc). Maybe these journals are lesser in traditional academic impact factor, but may have a higher true impact for the field of wildlife diseases if disseminated where the audience are wildlife disease managers.

How is the data managed in this WRMD resource? Is it shared with state and your federal/country wildlife disease managers? In terms of data access, is this data publicly available and are 'sensitive' locations anonymized in some way? That would be important if it were linked to public databases. Is the location of these outbreak events where animal was observed to be sick, or the location of the wildlife rehabilitation facility?

I suggest that the reviewers in their introduction actually mention the current wildlife disease surveillance systems in their country and Canada and what they do in terms of surveillance and what national surveillance programs are doing at least to provide some context and how these other systems may be failing in regards to early detection of wildlife mortality events through regional wildlife and resource managers and in some instances, hunters, that report observations of disease, and other surveillance programs like 'community science' approaches like finch mycoplasma disease surveys and monarch health surveys that are used for early detection, or EHD studies in deer, as well as feral hog disease studies. (I believe there are the national wildlife health center, Canadian cooperative wildlife health center, southeastern cooperative wildlife disease study). How do these wildlife surveillance networks interact. And, I think authors should to mention how their approach could improve these systems that are already In place, but you they mention specifically what these systems are in your country.

General questions- How do you account for mortality events of underrepresented species with low sample sizes with your methods? This approach may work for more common species, but I

think that wildlife mortality events in more threatened species may be overlooked and machine learning approaches may not be useful for these rare species due to the lack of sample size. Can you discuss this in the discussion briefly?

It appears that the WMME early alert system is particularly useful for avian/marine bird mortality. Perhaps you should be more specific in this article introduction that you mainly use data for avian mortality events as a test study system for better focus and strength of argument instead of just the term 'wildlife' because it can be misleading for readers?

Again, in the discussion, there is no evaluation of discussion of other wildlife surveillance methods currently in place in their region or country and specifically how the authors' system. What are the 'standard active systems (line 334)' in your country ?

Minor comments:

Line 79-81- The example of crow mortality as a syndromic event alerted by data from wildlife rehabilitation is not accurate. It was zoo staff and pathologists that noted an increase in dead crows around the zoo grounds. Furthermore, it was an astute veterinary pathologist with virology diagnostic laboratories that identified zoo deaths as WNV and then comparative evaluation of the etiologies of wild bird and human illness was performed. I recommend taking this example out since you are using it as an example of how wildlife rehabilitators facilitated early surveillance of WNV as again this is not accurate (lines 79-81).

Line 363- change 'wildlife' to 'avian'

Review form: Reviewer 3

Recommendation

Reject - article is scientifically unsound

Scientific importance: Is the manuscript an original and important contribution to its field?

Poor

General interest: Is the paper of sufficient general interest?

Acceptable

Quality of the paper: Is the overall quality of the paper suitable?

Marginal

Is the length of the paper justified?

Yes

Should the paper be seen by a specialist statistical reviewer?

Yes

Do you have any concerns about statistical analyses in this paper? If so, please specify them explicitly in your report.

No

It is a condition of publication that authors make their supporting data, code and materials available - either as supplementary material or hosted in an external repository. Please rate, if applicable, the supporting data on the following criteria.

Is it accessible?

Yes

Is it clear?

No

Is it adequate?

No

Do you have any ethical concerns with this paper?

No

Comments to the Author

Dear authors,

You have obviously done an extensive work with impressive results on networking and valuable definitions but I have a few major concerns that are described in the attached file. I encourage you to revise your manuscript accordingly and am confident that it could be published in another journal. What you report is certainly worth sharing.

Decision letter (RSPB-2020-2317.R0)

24-Nov-2020

Dear Dr Kelly:

I am writing to inform you that your manuscript RSPB-2020-2317 entitled "Early detection of wildlife morbidity and mortality through an event-based surveillance system" has, in its current form, been rejected for publication in Proceedings B.

This action has been taken on the advice of referees, who have recommended that substantial revisions are necessary. With this in mind we would be happy to consider a resubmission, provided the comments of the referees are fully addressed. However please note that this is not a provisional acceptance.

Sincerely,
 Professor Gary Carvalho
 mailto: proceedingsb@royalsociety.org

Associate Editor
 Board Member: 1

Comments to Author:

Following three expert reviews and my own review of the manuscript, there is a consensus that the manuscript submitted by Kelly et al. offers the potential to improve early detection of wildlife morbidity and mortality events by using pre-diagnostic data and real time processing using machine learning. Several reviewers express concerns that the authors have not sufficiently addressed the existing wildlife health surveillance literature nor fully articulated how the proposed surveillance system would be implemented. In considering any revision of their manuscript, the authors should please be sure to address each of the reviewer comments in detail as there were many potential issues raised and these will require the authors' careful attention.

Reviewer(s)' Comments to Author:

Referee: 1

Comments to the Author(s)

The authors present a proof of concept paper on use of wildlife rehabilitation centers as an early monitoring system for disease outbreaks. This can be a highly useful system for disease outbreaks and these centers are terribly underused for this purpose. There were minor edits that I made in the document, but my main concern is that the authors fail to demonstrate how truly important this system could be in detecting novel infectious or non-infectious pathogens. The examples in the paper utilizing existing pathogens on the landscape are important, but hardly demonstrate the usefulness of this system to an outside user. Epidemiologists would appreciate the opportunity to track existing diseases using this system, but the authors contend that it's also useful for novel pathogens. Thus, it would be nice if they could identify an example in the historic data set and include that example as well.

Referee: 2

Comments to the Author(s)

Main comments:

This is a article on a wildlife disease syndromic surveillance system using data from wildlife rehabilitators on a free online database that used machine learning methods to detect early wildlife mortality events in avian species. This WRMD seems to be a fantastic resource.

Your use and analyses of data on early detection of wild bird mortality events is important and novel for this field of wildlife diseases. The audience, which would be wildlife disease managers, state, provincial, and other types of government agencies dealing with wildlife surveillance, may miss this article in Proc B, but would more readily read the Journal of Wildlife Diseases, Wildlife Management, wildlife society journals, applied conservation journals, etc). Maybe these journals are lesser in traditional academic impact factor, but may have a higher true impact for the field of wildlife diseases if disseminated where the audience are wildlife disease managers.

How is the data managed in this WRMD resource? Is it shared with state and your federal/country wildlife disease managers? In terms of data access, is this data publicly available and are 'sensitive' locations anonymized in some way? That would be important if it were linked to public databases. Is the location of these outbreak events where animal was observed to be sick, or the location of the wildlife rehabilitation facility?

I suggest that the reviewers in their introduction actually mention the current wildlife disease surveillance systems in their country and Canada and what they do in terms of surveillance and what national surveillance programs are doing at least to provide some context and how these other systems may be failing in regards to early detection of wildlife mortality events through regional wildlife and resource managers and in some instances, hunters, that report observations of disease, and other surveillance programs like 'community science' approaches like finch mycoplasma disease surveys and monarch health surveys that are used for early detection, or EHD studies in deer, as well as feral hog disease studies. (I believe there are the national wildlife health center, Canadian cooperative wildlife health center, southeastern cooperative wildlife disease study). How do these wildlife surveillance networks interact. And, I think authors should to mention how their approach could improve these systems that are already in place, but you they mention specifically what these systems are in your country.

General questions- How do you account for mortality events of underrepresented species with low sample sizes with your methods? This approach may work for more common species, but I think that wildlife mortality events in more threatened species may be overlooked and machine learning approaches may not be useful for these rare species due to the lack of sample size. Can you discuss this in the discussion briefly?

It appears that the WMME early alert system is particularly useful for avian/marine bird mortality. Perhaps you should be more specific in this article introduction that you mainly use data for avian mortality events as a test study system for better focus and strength of argument instead of just the term 'wildlife' because it can be misleading for readers?

Again, in the discussion, there is no evaluation of discussion of other wildlife surveillance methods currently in place in their region or country and specifically how the authors' system. What are the 'standard active systems (line 334)' in your country ?

Minor comments:

Line 79-81- The example of crow mortality as a syndromic event alerted by data from wildlife rehabilitation is not accurate. It was zoo staff and pathologists that noted an increase in dead crows around the zoo grounds. Furthermore, it was an astute veterinary pathologist with virology diagnostic laboratories that identified zoo deaths as WNV and then comparative evaluation of the etiologies of wild bird and human illness was performed. I recommend taking this example out since you are using it as an example of how wildlife rehabilitators facilitated early surveillance of WNV as again this is not accurate (lines 79-81).

Line 363- change 'wildlife' to 'avian'

Referee: 3

Comments to the Author(s)

Dear authors,

You have obviously done an extensive work with impressive results on networking and valuable definitions but I have a few major concerns that are described in the attached file. I encourage you to revise your manuscript accordingly and am confident that it could be published in another journal. What you report is certainly worth sharing.

Author's Response to Decision Letter for (RSPB-2020-2317.R0)

See Appendix A.

RSPB-2021-0974.R0

Review form: Reviewer 2

Recommendation

Accept with minor revision (please list in comments)

Scientific importance: Is the manuscript an original and important contribution to its field?

Good

General interest: Is the paper of sufficient general interest?

Good

Quality of the paper: Is the overall quality of the paper suitable?

Good

Is the length of the paper justified?

Yes

Should the paper be seen by a specialist statistical reviewer?

Yes

Do you have any concerns about statistical analyses in this paper? If so, please specify them explicitly in your report.

No

It is a condition of publication that authors make their supporting data, code and materials available - either as supplementary material or hosted in an external repository. Please rate, if applicable, the supporting data on the following criteria.

Is it accessible?

Yes

Is it clear?

N/A

Is it adequate?

N/A

Do you have any ethical concerns with this paper?

No

Comments to the Author

Most comments of reviewers have been adequately addressed, However, the response to reviewer's 2 question if data publically available was not specifically addressed (it is useful to know- if it is not publically available that is fine, because some data might be protected for various reasons, but they should specify-I think they infer that data only available to those in the 'network?' is that rehabilitators and scientists, specifically).

Decision letter (RSPB-2021-0974.R0)

08-Jun-2021

Dear Dr Kelly

I am pleased to inform you that your manuscript RSPB-2021-0974 entitled "Early detection of wildlife morbidity and mortality through an event-based surveillance system" has been accepted for publication in Proceedings B.

The referee(s) have recommended publication, but also suggest some minor revisions to your manuscript. Therefore, I invite you to respond to the referee(s)' comments and revise your manuscript. Because the schedule for publication is very tight, it is a condition of publication that you submit the revised version of your manuscript within 7 days. If you do not think you will be able to meet this date please let us know.

Sincerely,
Professor Gary Carvalho
mailto: proceedingsb@royalsociety.org

Associate Editor
Comments to Author:

The authors have performed an extensive revision of their manuscript based on the considerable feedback they received in the previous round of review, and the manuscript is much improved as a result. Please be sure to address the remaining reviewer comments concerning data access as this is an important aspect of making the tools they developed here widely available.

Reviewer(s)' Comments to Author:

Referee: 2

Comments to the Author(s).

Most comments of reviewers have been adequately addressed, However, the response to reviewer's 2 question if data publically available was not specifically addressed (it is useful to know- if it is not publically available that is fine, because some data might be protected for various reasons, but they should specify-I think they infer that data only available to those in the 'network?' is that rehabilitators and scientists, specifically).

Author's Response to Decision Letter for (RSPB-2021-0974.R0)

See Appendix B.

Decision letter (RSPB-2021-0974.R1)

16-Jun-2021

Dear Dr Kelly

I am pleased to inform you that your manuscript entitled "Early detection of wildlife morbidity and mortality through an event-based surveillance system" has been accepted for publication in Proceedings B.

Data Accessibility section

Open Access

Paper charges

You are allowed to post any version of your manuscript on a personal website, repository or preprint server. However, the work remains under media embargo and you should not discuss it

with the press until the date of publication. Please visit <https://royalsociety.org/journals/ethics-policies/media-embargo> for more information.

Sincerely,
Proceedings B
<mailto:proceedingsb@royalsociety.org>

Appendix A

April 1, 2021

Editorial Board
Professor Gary Carvalho
Proceedings of the Royal Society B

Re: RSPB-2020-2317

Dear Professor Carvalho,

We are pleased to submit our revised manuscript, “Early detection of wildlife morbidity and mortality through an event-based surveillance system”- RSPB-2020-2317, for publication in the Proceedings of the Royal Society B. We are very grateful for the reviewers’ very thoughtful and thorough reviews of the paper and greatly appreciate the opportunity to address their comments in this improved version. We have revised the manuscript to address all comments and suggestions. The revisions substantially strengthened the manuscript and helped us to communicate our points more effectively.

We have responded point by point to the reviewers’ comments. The author responses are summarized in blue font.

Sincerely,

Terra Kelly

Dear Dr Kelly:

I am writing to inform you that your manuscript RSPB-2020-2317 entitled "Early detection of wildlife morbidity and mortality through an event-based surveillance system" has, in its current form, been rejected for publication in Proceedings B.

This action has been taken on the advice of referees, who have recommended that substantial revisions are necessary. With this in mind we would be happy to consider a resubmission, provided the comments of the referees are fully addressed. However please note that this is not a provisional acceptance.

Sincerely,

Professor Gary Carvalho

Associate Editor

Board Member: 1

Comments to Author:

Following three expert reviews and my own review of the manuscript, there is a consensus that the manuscript submitted by Kelly et al. offers the potential to improve early detection of wildlife morbidity and mortality events by using pre-diagnostic data and real time processing using machine learning. Several reviewers express concerns that the authors have not sufficiently addressed the existing wildlife health surveillance literature nor fully articulated how the proposed surveillance system would be implemented. In considering any revision of their manuscript, the authors should please be sure to address each of the reviewer comments in detail as there were many potential issues raised and these will require the authors' careful attention.

Reviewer(s)' Comments to Author:

Referee: 1

Comments to the Author(s)

The authors present a proof of concept paper on use of wildlife rehabilitation centers as an early monitoring system for disease outbreaks. This can be a highly useful system for disease outbreaks and these centers are terribly underused for this purpose. There were minor edits that I made in the document, but my main concern is that the authors fail to demonstrate how truly important this system could be in detecting novel infectious or non-infectious pathogens. The examples in the paper utilizing existing pathogens on the landscape are important, but hardly demonstrate the usefulness of this system to an outside user. Epidemiologists would appreciate the opportunity to track existing diseases using this system, but the authors contend that it's also useful for novel pathogens. Thus, it would be nice if they could identify an example in the historic data set and include that example as well.

Thank you, we greatly appreciate your thoughtful and thorough review of our paper. We have addressed each of your points in the paper and below.

Yes, great point, thank you. We made additions and edits throughout the paper to clearly demonstrate the system's utility for detecting anomalous events associated with an emerging disease (the first detection of a disease in the region) and highlight how the capacity to detect anomalies associated with emerging diseases illustrates the system's capacity for detecting anomalous events associated with a novel or previously unknown disease threat. Because the system detects increased admissions, it has the capacity to detect anomalies/events associated with diseases that fall into all the categories of common/endemic, emerging, and novel disease threats.

In the results section (Lines 274-294), we clarified that PPMV-1 and *Sarcocystis calchasi* are emerging pathogens in California and that detections of anomalous events in the system spurred in-depth investigations, which led to the identification of these emerging pathogens. Specifically, the text reads, “The WMME Alert System also alerted investigators to cases of neurologic disease in doves that were associated with the northward spread of Pigeon Paramyxovirus Type 1 (PPMV-1), an emerging virus in California.PCR and sequencing confirmed the presence of Pigeon Paramyxovirus-1, **the first detection of the virus emerging in Eurasian collared doves in this region of California** [46].

The WMME Alert System also detected increased admissions associated with an outbreak of neurologic disease in feral Rock Pigeons (*Columba livia*) in the San Francisco Bay area in late winter/early spring of 2017 (Figure 3b) [35]. Pan-*Sarcocystis* PCR identified *S. calchasi* group infections in several of the pigeons and sequences detected in eight of these cases had 100% homology with *S. calchasi* [35]. **This event demonstrated the emergence of this parasite in free-ranging Rock Pigeons in California** and highlighted the importance of increased surveillance in susceptible native columbids.”

In the discussion section (Lines 385-388), we added the following statement after discussing the range of threats the system was able to detect. “Detecting anomalies in admissions associated with emerging diseases in wildlife illustrates this system’s capacity to detect anomalous events associated with a novel threat.”

We also revised the column headings of Table 2 from “Common/endemic” and “Novel/emerging” to “Common/endemic threats” and “Emerging threats” to clarify that the events detected by the system were found to be associated with emerging pathogens.

We also incorporated all the reviewer’s minor edits that were made in the document.

Referee: 2

Comments to the Author(s)

Main comments:

This is an article on a wildlife disease syndromic surveillance system using data from wildlife rehabilitators on a free online database that used machine learning methods to detect early wildlife mortality events in avian species. This WRMD seems to be a fantastic resource.

Your use and analyses of data on early detection of wild bird mortality events is important and novel for this field of wildlife diseases. The audience, which would be wildlife disease managers, state, provincial, and other types of government agencies dealing with wildlife surveillance, may miss this article in Proc B, but would more readily read the Journal of Wildlife Diseases, Wildlife Management, wildlife society journals, applied conservation journals, etc). Maybe these

journals are lesser in traditional academic impact factor, but may have a higher true impact for the field of wildlife diseases if disseminated where the audience are wildlife disease managers.

Thank you, we greatly appreciate your thoughtful and thorough review of our paper. We have addressed each of your points in the paper and below. Several leading wildlife disease papers are now being published in the Proceedings of the Royal Society B, so our target audience of professionals in the fields of epidemiology, disease ecology, wildlife management and conservation, and biology reach for the Proceedings of the Royal Society B for literature. In addition, we have had interest in the system from professionals in the United Kingdom, Australia, and New Zealand, so we are interested in a journal that has broad international reach, like the Proceedings of the Royal Society B.

How is the data managed in this WRMD resource? Is it shared with state and your federal/country wildlife disease managers? In terms of data access, is this data publicly available and are 'sensitive' locations anonymized in some way? That would be important if it were linked to public databases. Is the location of these outbreak events where animal was observed to be sick, or the location of the wildlife rehabilitation facility?

Agreed, we added the following text to the paper for clarification on these points. "Data for each case includes a unique identifying case number, species, sex, age class, location where the animal was found, circumstances of admission, initial examination findings, and ultimately the diagnosis if determined. Personal identifiable information of rescuers is excluded. Aggregated data displayed through interactive tabular, graphical, and spatial dashboards in the WMME Alert System are accessible to the network and the Wildlife Health Laboratory (WHL) of the California Department of Fish and Wildlife (Rancho Cordova, CA), which leads the state's wildlife disease surveillance efforts."

I suggest that the reviewers in their introduction actually mention the current wildlife disease surveillance systems in their country and Canada and what they do in terms of surveillance and what national surveillance programs are doing at least to provide some context and how these other systems may be failing in regards to early detection of wildlife mortality events through regional wildlife and resource managers and in some instances, hunters, that report observations of disease, and other surveillance programs like 'community science' approaches like finch mycoplasma disease surveys and monarch health surveys that are used for early detection, or EHD studies in deer, as well as feral hog disease studies. (I believe there are the national wildlife health center, Canadian cooperative wildlife health center, southeastern cooperative wildlife disease study). How do these wildlife surveillance networks interact. And, I think authors should mention how their approach could improve these systems that are already in place, but you they mention specifically what these systems are in your country.

Great suggestion, thank you. We made substantial additions and edits to the text in the introduction and discussion sections (as suggested below) to include information on the wildlife

disease surveillance systems/programs in the U.S. and Canada, what types of information they provide in terms of a disease surveillance strategy, interactions between systems, and what valuable information the WMME Alert System brings for a more comprehensive wildlife disease surveillance strategy.

For instance, in the introduction section (Lines 87-100), we included the following details on surveillance systems:

“In North America, wildlife agencies conduct targeted disease surveillance for several endemic and recently emerging wildlife diseases of importance (e.g., avian influenza, rabies, white-nose syndrome, snake fungal disease, and bovine tuberculosis) [27–30]. These agencies also investigate reports of unusual wildlife morbidity and mortality events in order to detect anomalies outside of active surveillance targets, including new species or geographic areas affected by known diseases or new diseases as they emerge.

With increasing focus on the importance of early detection and the need for innovative, cost-effective surveillance strategies, several new approaches and tools have been developed as a complement to conventional surveillance systems. For example, citizen science has increasingly been used in North America for surveilling diseases causing characteristic clinical signs, such as avian pox or house finch conjunctivitis [20,31]. Agencies have also implemented harvest-based disease surveillance as a practical, cost-effective strategy (e.g., chronic wasting disease in cervids and *Brucella* spp. in coyotes) [32,33].

Wildlife rehabilitation organizations are also increasingly recognized for their potential to contribute to disease surveillance, including for diseases of importance to domestic animal and human health [34–36] as well as emerging threats [31,34,37].When linked through a formal network, these organizations have the potential to assimilate and share large amounts of clinical data in near real-time, collectively representing a highly valuable and under-utilized resource that can be used to complement existing surveillance efforts for early and enhanced detection [21,31,34,35,38–42].”

General questions- How do you account for mortality events of underrepresented species with low sample sizes with your methods? This approach may work for more common species, but I think that wildlife mortality events in more threatened species may be overlooked and machine learning approaches may not be useful for these rare species due to the lack of sample size. Can you discuss this in the discussion briefly?

Yes, this is an important point regarding the performance of our methods (machine learning clinical classification model and statistics to detect anomalous patterns) in addressing wildlife mortality in species with small sample sizes. We have included text in the results and discussion sections to address this question and have included some additional clarification on the nuances below.

The machine learning clinical classification model (the model classifying cases according to the clinical classifications) is not affected by the rarity of species as the model is independent of the species and instead is based on the clinical data entered by the rehabilitator or clinician (reasons for admission and physical examination findings). The statistical methods for identifying anomalies in the data identified trends and anomalies in rare species, including species for which only 100 individuals were admitted to the network over the five-year period (See Figure 1 below for Osprey).

Figure 1: Osprey admitted to the wildlife rehabilitation network from 2013-2018 (n = 113). The blue line shows the raw data and the black line shows the moving average of admissions. The shaded area represents two times the moving standard deviation. The red dots indicate anomalies when more than the expected number of cases are admitted.

Some of the strengths of this system include its capacity to detect anomalies or wildlife morbidity events using a pre-diagnostic/ syndromic approach and flexibility to demonstrate patterns in individual species as well as groups of species (i.e., taxa). Although, the system's specificity is lower for identifying trends for a single rare species/clinical classification combination given small sample size (see Figure 2 below for cases of neurologic disease in Golden Eagles), the system is sensitive to and will detect anomalies, which may result in alerts with a single or few cases that require additional scrutiny and follow-up to determine whether an investigation is warranted. WMME Alert System users can also investigate trends at the taxonomic family level inclusive of the rare species to investigate threats across related species.

Figure 2: Trend in admissions of neurologic disease cases in Golden Eagles across the wildlife rehabilitation network from 2013-2018 ($n = 5$). The blue line shows the raw data. The black line shows the moving average and the shaded area represents two times the moving standard deviation. The red dots indicate anomalies when more than the expected number of cases are reported.

In the results section, we included the following text (Lines 317-323): “Events in rare species, such as increased cases of neurologic disease in Golden Eagles (*Aquila chrysaetos*) (Figure S2), can also be monitored in the system. However, the specificity for alerts in these species tends to be lower given the relatively fewer numbers presenting to organizations. Exploring trends at the taxonomic family level (i.e. *Accipitridae* family), in addition to the species level, could provide additional insights into the health of threatened and rare species (Figure 4h).”

Figure 3: Trend of admissions of neurologic disease cases in birds from the *Accipitridae* family (birds of prey) in the participating wildlife rehabilitation centers across California from 2013-2018. The blue line shows the raw data and the black line shows the moving average. The shaded area represents two times the moving standard deviation and the red dots indicate anomalies when more than the expected number of cases are reported.

We also included the following text in the discussion section (Lines 402-408), “In addition, even though the system’s specificity was lower for detecting events in rare species, alerts involving a small number of individuals of a threatened or endangered species signifying a potential anomalous event may be worthy of investigation. Monitoring for alerts in sympatric species and/or in related but more common species at a taxa level might also cue investigators into a common threat that could impact the health of threatened and endangered species. The precision of the model will also improve over time as the system becomes more populated with data.”

It appears that the WMME early alert system is particularly useful for avian/marine bird mortality. Perhaps you should be more specific in this article introduction that you mainly use data for avian mortality events as a test study system for better focus and strength of argument instead of just the term ‘wildlife’ because it can be misleading for readers?

Agreed, we added more examples of how the system alerted to anomalies in mammalian admissions and included the following text in the results section of the paper (Lines 312-317): “Various investigations in mammals were also triggered due to system alerts. For example, canine distemper virus (CDV) infection and bromethelin intoxication were associated with anomalies in raccoon and skunk admissions (Figure 4f). Not surprisingly, the system was also able to track trends in admissions of animals associated with physical injury (e.g., vehicular trauma in deer; Figure 4g), a common circumstance of admission to rehabilitation organizations.”

Again, in the discussion, there is no evaluation of discussion of other wildlife surveillance methods currently in place in their region or country and specifically how the authors’ system. What are the ‘standard active systems (line 334)’ in your country?

In addition to the material that was added to the introduction section, we included the following text in the discussion section: “However, this system, together with other general disease surveillance efforts (i.e., citizen wildlife mortality event reporting), are important complements to targeted surveillance efforts in California (e.g., chronic wasting disease in cervids and white nose syndrome in bats) through efficient monitoring for emerging threats across a broad range of species and geographies [40], especially species in disturbed environments [35]. The information generated through this system adds value to other general surveillance strategies through its ability to rapidly and efficiently detect threats that lead to illness and death in wild animals but do not necessarily result in conspicuous mortality events that would be detected through citizen reporting streams.”

Minor comments:

Line 79-81- The example of crow mortality as a syndromic event alerted by data from wildlife rehabilitation is not accurate. It was zoo staff and pathologists that noted an increase in dead crows around the zoo grounds. Furthermore, it was an astute veterinary pathologist with virology

diagnostic laboratories that identified zoo deaths as WNV and then comparative evaluation of the etiologies of wild bird and human illness was performed. I recommend taking this example out since you are using it as an example of how wildlife rehabilitators facilitated early surveillance of WNV as again this is not accurate (lines 79-81).

Agreed, we clarified the point we were trying to make in this section. It now reads “With increasing awareness of the impacts of emerging diseases on both humans and animals and the importance of wild animals as reservoirs of zoonoses, there is greater recognition of the need for disease surveillance in free-ranging wildlife. Reports of unusual illnesses or deaths in wildlife populations may serve as the first alert of an emerging health threat [18]. For example, mortality in crows (*Corvus* species) and exotic birds at a zoologic park in New York provided early warning of the emergence of West Nile Virus (WNV) in the U.S. in 1999. Avian deaths were a critical precursor to identifying WNV as the cause of the associated human encephalitis outbreak [19]. As crow mortalities were occurring prior to the onset of human WNV cases, dead bird surveillance served as a sensitive indicator for WNV activity and risk of infection for humans [20].”

Line 363- change ‘wildlife’ to ‘avian’

We expanded our results to include several examples in which the system detected anomalous admissions/events in mammalian species. In lines 312-317 of the results section, the paper now reads “Various investigations in mammals were also triggered due to system alerts. For example, canine distemper virus (CDV) infection and bromethelin intoxication were associated with anomalies in raccoon and skunk admissions (Figure 4c). Not surprisingly, the system was also able to track trends in admissions of animals associated with physical injury (e.g., vehicular trauma in deer; Figure 4d), a common circumstance of admission to rehabilitation organizations.”

Referee: 3

Comments to the Author(s)

Dear authors,

You have obviously done extensive work with impressive results on networking and valuable definitions but I have a few major concerns that are described in the attached file. I encourage you to revise your manuscript accordingly and am confident that it could be published in another journal. What you report is certainly worth sharing.

Thank you, we greatly appreciate your thoughtful and thorough review of our paper. We have addressed each of your points in the paper and below.

Attached file

Early detection of wildlife morbidity and mortality through an event-based surveillance system by Terra et al.

This manuscript reports statistical efforts to develop an algorithm able to detect morbidity and mortality events as an early warning system for health events affecting wildlife. For this purpose the authors have used data from wildlife care centers across California. They have done an impressive work, including a admirable effort on network development and definitions of syndromes, as well an extensive statistical work. However, I have major concerns regarding the originality of the work and the lack of consideration of previous published studies and known surveillance concepts.

General surveillance systems (as opposed to targeted/specific surveillance) usually rely on postmortem investigations and the potential contribution of wildlife care centers to wildlife health surveillance has been repeatedly mentioned in the scientific literature. What is new in the present study is the structured network of such centers and the obvious organisation towards a serious role in surveillance, as well as the harmonization of data collection and data sharing. The demonstration of the potential of such an organisation is worth to be underlined but needs to be better shown in the paper. Importantly, general surveillance should not be mentioned as a novel approach. Rather, the contribution of care centers should be seen as a complement to already existing systems. Alternatively, they could play a pioneer role in regions of the world where care centers exist while there is a lack of structures and/or competencies for collecting carcasses and carrying out wildlife necropsies.

Thank you, we have made substantial additions and edits to the introduction and discussion sections for clarification to address these concerns. We added the following text to lines 66-86 in the introduction section to clarify the unique contribution this type of general disease surveillance system adds to existing disease surveillance programs, including other general disease surveillance strategies.

“Therefore, wildlife disease surveillance programs typically also incorporate general disease surveillance, a strategy for detecting sick and dead animals in the wild and identifying causes of morbidity and mortality [23]. General disease surveillance is not limited to a few species or pathogens, rather it covers a broad range of wildlife and causes of illness and death [23].

General surveillance systems, especially systems that utilize existing pre-diagnostic health information or syndromic data can facilitate early detection of health threats through identification of unusual disease clusters early before diagnoses are confirmed and officially reported [18,24]. While syndromic systems do not track verified events, they can provide a valuable complement to targeted surveillance through alerting to anomalous wildlife morbidity events, thereby enhancing situational awareness and increasing opportunities for early detection and response. Syndromic classification of wildlife mortalities based on pre-diagnostic postmortem examination findings has been highlighted as a rapid, reliable, and relatively inexpensive strategy for disease surveillance [25]. Utilizing pre-diagnostic clinical wildlife health data generated through physical examination findings, is a novel strategy that offers an

even more efficient approach to syndromic surveillance as the data is entered in near real-time upon admission of animals to the centers. It is also practical in settings where it is not feasible to classify high numbers of cases based on their pathologic profiles through postmortem examinations. By leveraging existing clinical wildlife health data, this approach can also provide a relatively inexpensive means to bolster conventional wildlife disease surveillance programs [26].”

We also added the following text in lines 364-368 of the discussion section to further clarify the additional value this type of system brings as a complement to other surveillance streams for a comprehensive disease surveillance program incorporating a variety of targeted and general disease surveillance efforts, “The information generated through this system adds value to other general surveillance strategies through its ability to rapidly and efficiently detect threats that lead to illness and death in wild animals but do not necessarily result in conspicuous mortality events that would be detected through citizen reporting streams.”

Specific comments:

Introduction

This section needs improvement.

First, the introduction of the topic should be better structured:

- L36-39 introduces the reader to the problematic of environmental degradation affecting wildlife
- L40-44 jump to the problematic of zoonoses. According to the following text, L40 «The increasing incidence of EIDs...» must refer to human diseases but the previous text was suggesting that the issue to be looked at would be diseases affecting wildlife.
- The next paragraph starts with wildlife as a source of human pathogens, which would be best placed directly after L40-44, and continues with concerns about biodiversity loss.
- L53 «threats» are mentioned. I suggest to restructure the beginning of the introduction (currently the two threats are a little mixed up), separating and clarifying the two types of threats arising from environmental alterations and changes in human behavior and the resulting increased interactions at the wildlife/domestic animal/wildlife interface, i.e. (1) impact on wildlife health (2) increased risk of zoonotic transmission of infectious agents originating from wildlife

Agreed, the introduction has been restructured to separate and clarify the major threats (threats to wildlife health and increased risk of transmission of wildlife-borne zoonotic pathogens to humans and domestic animals) in line with your guidance.

Second, and this is a major concern with this paper, the authors do not seem to be aware that specific (or targeted/active) surveillance is only one component of health surveillance systems. The basis of health surveillance is general/passive surveillance and a proper wildlife health surveillance program should include both approaches, which are complementary, and this is not a new concept at all. General surveillance is crucial for the detection of pathogen-unspecific health events (incl. intoxication and adverse environmental conditions). See for example the Training Manual of the OIE about wildlife disease surveillance (https://www.oie.int/fileadmin/Home/eng/International_Standard_Setting/docs/pdf/WGWildlife/A_Training_Manual_Wildlife_2.pdf) or p. 15 of the Terrestrial Animal Code (<https://www.oie.int/en/standard-setting/terrestrial-code/access-online/>) or Kuiken et al. 2011 Rev. sci. tech. Off. int. Epiz., 2011, 30 (3), 755-761.

We have made substantial additions and revisions throughout the paper to clarify that the WMME Alert System uses a general surveillance strategy that is complementary to other general and targeted disease surveillance efforts in California's wildlife disease surveillance program. We have added the OIE reference you included as an example in the introduction when discussing the different components of disease surveillance systems and the value that the WMME Alert System brings to the general disease surveillance efforts. In the introduction and discussion sections, we added details to clarify the novelty and added value of a system that uses existing pre-diagnostic clinical data to rapidly and efficiently detect threats that do not necessarily result in conspicuous mortality events that would be detected through existing general disease surveillance efforts (citizens reporting mortality events).

“Surveillance strategies focusing on early detection are important for quickly identifying and mitigating threats as they emerge [18]. Targeted surveillance focusing on a particular pathogen or contaminant is valuable for surveilling specific disease agents in wildlife [21–23]. However, implementation of targeted surveillance is costly and time-consuming for surveilling a wide range of threats across multiple species [20,21,23]. Targeted surveillance strategies can also miss trends signifying an emerging threat that is outside the systems' targets. Therefore, wildlife disease surveillance programs typically also incorporate general disease surveillance, a strategy for detecting sick and dead animals in the wild and identifying causes of morbidity and mortality [23]. General disease surveillance is not limited to a few species or pathogens, rather it covers a broad range of wildlife and causes of illness and death [23].

General surveillance systems, especially systems that utilize existing pre-diagnostic health information or syndromic data can facilitate early detection of health threats through identification of unusual disease clusters early before diagnoses are confirmed and officially reported [18,24]. While syndromic systems do not track verified events, they can provide a valuable complement to traditional surveillance through alerting to anomalous events causing wildlife morbidity, thereby enhancing situational awareness and increasing opportunities for early detection and response. Syndromic classification of wildlife mortalities based on pre-diagnostic postmortem examination findings has been highlighted as a rapid, reliable, and

relatively inexpensive strategy for disease surveillance [25]. Utilizing pre-diagnostic clinical wildlife health data generated through physical examination findings, is a novel strategy that offers an even more efficient approach to syndromic surveillance as the data is entered in near real-time upon admission of animals to the centers. It is also practical in settings where it is not feasible to classify high numbers of cases based on their pathologic profiles through postmortem examinations. By leveraging existing clinical wildlife health data, this approach can also provide a relatively inexpensive means to bolster disease surveillance programs [26].“

Furthermore, the application of syndromic surveillance to wildlife to improve early warning in a surveillance program based on a network of institutions diagnosing on wildlife disease was already described by other authors (<https://bmcvetres.biomedcentral.com/articles/10.1186/1746-6148-6-56>), which used necropsy data to apply algorithms for early detection of abnormal health events. I am not qualified to judge the obviously extensive statistic work carried out by the authors, and to compare algorithms used in the two studies but previous work should be mentioned as well as any advantage of the newly developed detection system compared to previous published efforts.

We agree and thank you for highlighting the importance of including discussion of this paper. A key difference between the two approaches is that the system described by Warns-Petit et al. is based on post-mortem examination data from dead animals, which is different from our system which is based on clinical data entered upon admission of the animals to the centers. Using clinical data (versus post-mortem examination data) has a few advantages in that it is a highly efficient approach to syndromic surveillance as data is entered in near real-time upon admission of animals and it is practical in settings in which it is not logistically or financially feasible to classify cases based on their pathologic profiles through postmortem examinations. We added the following text to lines 78-88 of the introduction, “Syndromic classification of wildlife mortalities based on pre-diagnostic postmortem examination findings has been highlighted as a rapid, reliable, and relatively inexpensive strategy for disease surveillance [25]. Utilizing pre-diagnostic clinical wildlife health data generated through physical examination findings, is a novel strategy that offers an even more efficient approach to syndromic surveillance as the data is entered in near real-time upon admission of animals to the centers. It is also practical in settings where it is not feasible to classify high numbers of cases based on their pathologic profiles through postmortem examinations. By leveraging existing clinical wildlife health data, this approach can also provide a relatively inexpensive means to bolster disease surveillance programs [26].”

At the end of the introduction, the sentence «Here, we discuss the development and pilot of an online surveillance tool that integrates pre-diagnostic data entered in near real-time by a network of wildlife rehabilitation organizations to enable early and enhanced detection of wildlife morbidity and mortality events.» should be rewritten to present the goal of the study, e.g. «In this study we aimed to develop...»

Thank you for the suggestion. This sentence now reads, “In this study, we aimed to develop and pilot an online surveillance system that integrates pre-diagnostic clinical health data entered in near real-time by a network of wildlife rehabilitation organizations to facilitate early and enhanced detection of wildlife morbidity and mortality events.”

Methods & Results:

The clinical data are put forward (logically if the data originate from wildlife care centers) but postmortem analyzes are also mentioned. The authors should clarify which kind of data were used for their early detection analysis (clinical and/or post-mortem data) and how the two types of data are linked in their material. For instance, in the mentioned examples, I don't understand if the developed early detection system allowed to predict mortality events based on clinical data collected in care centers (what was detected with which type of data?).

Thanks for the suggestion. Throughout the paper, we added text to clarify that system detected anomalies based on the pre-diagnostic clinical data (based on reasons for admission and physical examination findings) that was entered in near real-time upon admission to the wildlife rehabilitation organizations.

For instance in the introduction section lines 80-85 now read as follows: “ Utilizing pre-diagnostic clinical wildlife health data generated through physical examination findings, is a novel strategy that offers an even more efficient approach to syndromic surveillance as the data is entered in near real-time upon admission of animals to the centers. It is also practical in settings where it is not feasible to classify high numbers of cases based on their pathologic profiles through postmortem examinations. By leveraging existing clinical wildlife health data, this approach can also provide a relatively inexpensive means to bolster disease surveillance programs [26].”

In addition, lines 108-11 were revised to read, “In addition, efforts to utilize wildlife rehabilitation organization data to contribute to wildlife disease surveillance have previously focused on verified diagnostic data rather than clinical pre-diagnostic data limiting the information's utility to contribute to early detection.” Another example, is the clarifying text in lines 139-142, which now read “Alerts to anomalous events are generated when the number of cases exceeds pre-defined thresholds for the number of admissions for a given species and for the number of admissions for a given species presenting with a specific clinical classification (e.g., neurologic disease) based on physical examination findings.”

See also Discussion L301-303.

This sentence has been revised. It now reads “We demonstrate the use of an online surveillance system integrating clinical pre-diagnostic data from a network of wildlife rehabilitation organizations to facilitate early and enhanced detection of wildlife morbidity and mortality events in California.”

Discussion:

L304-311 + 341-343 These situations are inherent in any general surveillance program, whether based on clinical or post-mortem data.

We have added some context to this section for clarification. The paragraph now reads, “This finding reflects the inherent reporting bias of wildlife disease surveillance systems that rely on the public for initial detection of cases. However, this system, together with other general disease surveillance efforts (i.e., citizen wildlife mortality event reporting), are important complements to targeted surveillance efforts in California (e.g., chronic wasting disease in cervids and white nose syndrome in bats) through efficient monitoring for emerging threats across a broad range of species and geographies [40], especially species in disturbed environments [35]. The information generated through this system adds value to other general surveillance strategies through its ability to rapidly and efficiently detect threats that lead to illness and death in wild animals but do not necessarily result in conspicuous mortality events that would be detected through citizen reporting streams.”

L319-322 This is not specific to care centers.

Agreed, we revised this paragraph to the follow to clarify the point we are trying to make: “As front-line responders for injured and sick wild animals, wildlife rehabilitation organizations are well poised to detect index cases of emerging wildlife health threats [34,36]. Enhanced capacity to quickly identify unusual cases or patterns is becoming more important with increasing anthropogenic pressures causing unforeseen threats (emerging infectious diseases, and environmental pollutants) that can result in population declines [48] and endangerment of common species [49]. As emerging threats become more commonplace, there is a greater need for wildlife disease surveillance programs that extend beyond tracking only known hazards [50] and have the capacity to rapidly detect small isolated events.”

L323-327 I am afraid it was the veterinary examinations that detected the diseases, not the system. The system value rather consisted in the collection of information at the scale of the State rather than at the scale of single centers and thus to analyse and generate information for a large geographical area. This should be better explained.

Agreed, this section has been revised to clarify that the system picks up on anomalous events in the data that trigger alerts for in-depth disease investigations. It now reads as follows, “This surveillance application was effective in detecting anomalous patterns of admissions across the network of organizations that upon investigation were determined to be the result of both common and emerging health threats. Common health threats such as *Mycoplasma* spp. conjunctivitis in songbirds, trichomoniasis in doves, CDV in raccoons, and petroleum contamination of marine birds were detected with support from this system’s alerts, illustrating its utility for monitoring trends in these diseases over time. The system also detected events that

upon investigation were identified to be the result of emerging diseases in peridomestic and/or invasive species that present a threat to native wildlife [35,46]. Detecting anomalies in admissions associated with emerging diseases in wildlife illustrates this system's capacity to detect anomalous events associated with a novel threat."

L331-340 I may have missed something while reading the manuscript but I don't understand how the system contributes to early warning: does it allow to implement prevention measures to protect animal and/or human health? From the provided examples I rather understand that the system allowed to document unusual disease events at a large geographical scale.

This section has been substantially revised and no longer refers to early warning. Text throughout the manuscript refers to the system's value in early detection of morbidity and mortality events. Early detection and investigation of an event in wildlife that has implications for animal and human health could allow for implementation of prevention and control measures to protect animal and human health.

Others: The varying accuracy of the model as concerns the clinical classification of cases and its potential impacts on effective surveillance and reliability of the alerts should be discussed.

Agreed, we included text discussing the varying accuracy of the clinical classification model. This section now reads, "Overall, the model used to predict the clinical classifications demonstrated high accuracy. Misclassification of cases occurred primarily due to similarities in vocabulary in the reasons for admission and initial physical examination fields across multiple classifications. For example, birds presenting with physical injury and head trauma were sometimes misclassified as neurological disease cases due to similar verbiage across those two classifications. This type of misclassification can be reduced through inclusion of multiple clinical classifications. A multi-output classification system, in which a single case can be assigned two or more clinical classifications, is currently under development in the system. In addition, even though the system's specificity was lower for detecting events in rare species, alerts involving a small number of individuals of a threatened or endangered species signifying a potential anomalous event may be worthy of investigation. Monitoring for alerts in sympatric species and/or in related but more common species at a taxa level might also cue investigators into a common threat that could impact the health of threatened and endangered species. The precision of the model will also improve over time as the system becomes more populated with data."

Conclusion:

The work would be of interest for a wider audience if the interest of the system would be more generalised and not focused on California. This large State should rather be regarded as an example. The last sentence goes into this direction but the whole Conclusion could be formulated

in this way. In the introduction, the authors mention two threats. It would be nice to «close the loop» and explain how the surveillance system addresses these threats.

Agreed, nice suggestion. We have included the following text in the conclusion section: “We provide proof of concept for utilizing pre-diagnostic clinical data assimilated from a network of wildlife rehabilitation organizations to contribute to early and enhanced detection of wildlife morbidity and mortality events. The WMME Alert System serves as a model for a relatively efficient, inexpensive system that capitalizes on existing data sources to augment existing surveillance and monitoring efforts and promote situational awareness. In addition, the platform or its framework provides an effective strategy for early detection of anomalous events across broad species, geographies, and threats and has the capacity to be scaled up, adapted, and applied in other regions or contexts, including those where diagnostic capacity is limited. It serves as a valuable tool for assisting with early detection of and alerting to emerging diseases of wildlife as well as threats to domestic animal and human health (e.g., harmful algal blooms). The potential exists to expand the network to additional organizations involved in wildlife care and research and to create separate networks in other regions around the world given the current reach of WRMD.”

Appendix B

June 11, 2021

Editorial Board
Professor Gary Carvalho
Proceedings of the Royal Society B

Re: RSPB-2021-0974

Dear Professor Carvalho,

We are pleased to submit our revised manuscript, “Early detection of wildlife morbidity and mortality through an event-based surveillance system”- RSPB-2021-0974, for publication in the Proceedings of the Royal Society B. We appreciate the opportunity for a second review and have revised the manuscript to address the additional comment. The author response is summarized in blue font.

Thank you,

Terra Kelly

Reviewer(s)' Comments to Author:

Referee: 2

Comments to the Author(s).

Most comments of reviewers have been adequately addressed, However, the response to reviewer's 2 question if data publically available was not specifically addressed (it is useful to know- if it is not publically available that is fine, because some data might be protected for various reasons, but they should specify-I think they infer that data only available to those in the 'network?' is that rehabilitators and scientists, specifically).

Currently, the WMME Alert System is only available to wildlife rehabilitation centers within the network and the California Department of Wildlife. We have revised the text in the manuscript to make that clearer. It now reads, "Aggregated data displayed through interactive tabular, graphical, and spatial dashboards in the WMME Alert System are accessible to the network of partner wildlife rehabilitation organizations and the Wildlife Health Laboratory (WHL) of the California Department of Fish and Wildlife, which leads the state's wildlife disease surveillance efforts."